# CIRCUIT COMPLEXITY BOUNDS FOR VISUAL AUTOREGRESSIVE MODEL

## ABSTRACT

Understanding the expressive ability of a specific model is essential for grasping its capacity limitations. A recent breakthrough in image generation is the introduction of Visual Autoregressive (VAR) Models, which employ a scalable coarse-to-fine "next-scale prediction" framework. We investigate the circuit complexity of the VAR model and establish a bound in this study. Our primary result demonstrates that the VAR model is equivalent to a simulation by a uniform $\mathsf{TC}^0$ threshold circuit with hidden dimension $d$ and $\mathrm{poly}(d)$ precision. This is the first study to rigorously highlight the limitations in the expressive power of VAR models despite their impressive performance. We believe our findings will offer valuable insights into the inherent constraints of these models and guide the development of more efficient and expressive architectures in the future.

## 1 INTRODUCTION

Visual generation has seen widespread applications across various domains, including image restoration (Lin et al., 2025; Guo et al., 2025), augmented reality (Azad et al., 2024b), medical imaging (Azad et al., 2024a; Ma et al., 2024; Li et al., 2025), and creative industries such as game development (Rafner et al., 2020; Chen et al., 2025a). By generating realistic and diverse images from textual descriptions or other forms of input, visual generation models are transforming how machines perceive and produce visual content. Among the most popular models for visual generation are Variational AutoEncoders (VAE) (Doersch, 2016), Generative Adversarial Networks (GAN) (Goodfellow et al., 2020), Diffusion models (Sohl-Dickstein et al., 2015; Ho et al., 2020), and Flow-based models (Kingma & Dhariwal, 2018). These models have made notable progress in producing high-quality, high-resolution, and diverse images, expanding the potential of visual generation through improvements in realism, diversity, and fidelity.

However, the introduction of the Visual AutoRegressive model (VAR) (Tian et al., 2024) represents a significant shift in the paradigm in this field. Instead of the traditional "next-token prediction", the VAR model adopts a coarse-to-fine "next-scale prediction" approach. Through this innovative approach, the VAR model is able to capture visual distributions more effectively, exceeding the performance of diffusion transformers in image generation tasks. Additionally, VAR's zero-shot generalization capability spans multiple tasks, including image inpainting and manipulation. These results suggest that VAR offers a promising direction for autoregressive models in visual generation.

As the VAR model demonstrates its impressive performance, it is crucial to explore the limitations of the expressiveness of the VAR model. Up to now, the expressiveness from a circuit complexity perspective of the VAR model remains underexplored. This gap raises an important question:

*What are the limitations of the expressive power of the VAR model in terms of circuit complexity?*

To explore this issue, we apply circuit complexity theory, which offers valuable tools for analyzing the computational resources needed for specific tasks. By representing the VAR model as complexity circuits, we can systematically evaluate their capabilities and determine the lower bounds of the problems they can address.

In this work, we present a comprehensive theoretical investigation into the circuit complexity bounds of the VAR models. Our approach involves analyzing and formulating the architecture of the

VAR model and analyzing the computational complexity of its components, such as pyramid up-interpolation layers, convolution layers and transformer layers, etc. Finally, we show that uniform $\mathsf{TC}^0$ circuits can efficiently simulate these models.

The primary contributions of our work are summarized below:

- As far as we know, this is the first paper to present a mathematical formulation of the Visual AutoRegressive model (Section 4).

- We prove that the VAR model with $\mathrm{poly}(d)$-precision, $O(1)$ depth and $\mathrm{poly}(d)$ size can be simulated by a DLOGTIME-uniform $\mathsf{TC}^0$ circuit family (Theorem 5.13).

**Roadmap.** Section 2 offers a summary of the related works. Section 3 introduces the necessary notations and definitions for the subsequent analysis. In Section 4, we present the mathematical formulation of the VAR model. Section 5 details the circuit complexity results for the VAR model. Section 6 presents the conclusions of our work.

## 2 RELATED WORK

### 2.1 CIRCUIT COMPLEXITY AND NEURAL NETWORK

In computational theory, circuit complexity (Arora & Barak, 2009) refers to the classification and analysis of computational problems based on the size and depth of Boolean circuits required to solve them, aiming to understand the inherent difficulty of problems in terms of circuit resources. Circuit Complexity has important applications in understanding the capabilities of deep learning models (Pérez et al., 2019; Hahn, 2020; Liu et al., 2022; Hao et al., 2022; Merrill et al., 2022; Merrill & Sabharwal, 2023; Feng et al., 2024; Chen et al., 2025b; Li et al., 2024a; Chen et al., 2024; Li et al., 2024b). Specifically, (Hahn, 2020) investigates the computational boundaries of self-attention, demonstrating that, despite its effectiveness in NLP tasks, it has difficulty modeling periodic finite-state languages and hierarchical structures without scaling up the number of layers or attention heads. (Feng et al., 2024) delves into the theoretical underpinnings of Chain-of-Thought (CoT) within LLMs, demonstrating its ability to solve complex tasks like arithmetic and dynamic programming through sequential reasoning process, despite the limitations of bounded-depth Transformers. Recently, (Chen et al., 2025b) shows that Mamba and State-space Models (SSMs) have the same computational limits as Transformers, residing within the DLOGTIME-uniform $\mathsf{TC}^0$ complexity class. To the best of our knowledge, circuit complexity theory has not yet been used to analyze the computational constraints of Visual AutoRegressive models.

### 2.2 LIMITATION OF TRANSFORMER ARCHITECTURE

Transformer Architecture has shown remarkable success in various fields, particularly in natural language processing, reinforcement learning, and computer vision. By leveraging self-attention mechanisms to capture long-range dependencies, the Transformer has become the architecture of choice for applications such as machine translation (Raganato & Tiedemann, 2018; Wang et al., 2019; Yao & Wan, 2020) and image generation (Parmar et al., 2018; Ding et al., 2021; Tian et al., 2024). Recently, a series of studies have shed insight into the reasoning limitations of Transformer Architecture (Merrill et al., 2022; Merrill & Sabharwal, 2023; Feng et al., 2024; Merrill & Sabharwal, 2024; Liang et al., 2025; Ke et al.; Chiang, 2024). Specifically, (Merrill et al., 2022) showed that a generalized form of hard attention can recognize languages that go beyond what the $\mathsf{AC}^0$ class can compute, with the $\mathsf{TC}^0$ class serving as an upper bound for the formal languages it can identify. The study by (Liu et al., 2022) established that softmax-transformers (SMATs) are included in the non-uniform $\mathsf{TC}^0$ class. As a next step, (Merrill & Sabharwal, 2023) demonstrated that SMATs belong to L-uniform $\mathsf{TC}^0$ class. Recently, (Chiang, 2024) demonstrated that average-hard attention transformers (AHATs), without approximation, and SMATs with floating-point precision of $O(\mathrm{poly}(n))$ bits, as well as SMATs with at most $2^{-O(\mathrm{poly}(n))}$ absolute error, can all be classified in the DLOGTIME-uniform $\mathsf{TC}^0$ class.

## 3 PRELIMINARY

Section 3.1 explains the basics of circuit complexity classes. Section 3.2 introduces key simulations of floating-point operations, which will be used in later sections for the proofs.

**Notations.** We apply $[n]$ to represent the set $\{1, 2, \cdots, n\}$ for any positive integer $n$. The set of natural numbers is denoted by $\mathbb{N} := \{0, 1, 2, \ldots\}$. Let $X \in \mathbb{R}^{m \times n}$ be a matrix, where $X_{i,j}$ refers to the element at the $i$-th row and $j$-th column. When $x_i$ belongs to $\{0, 1\}^*$, it signifies a binary number with arbitrary length. In a general setting, $x_i$ represents a length $p$ binary string, with each bit taking a value of either 0 or 1.

### 3.1 KEY CONCEPTS IN CIRCUIT COMPLEXITY

We discuss several circuit complexity classes, starting with the concept of a boolean circuit.

**Definition 3.1** (Boolean Circuit, Definition 6.1 in (Arora & Barak, 2009)). *A Boolean circuit with input size $d$, where $d \in \mathbb{N}$, corresponding to a function that $C_d : \{0, 1\}^d \to \{0, 1\}$. This circuit can be typically represented as a directed acyclic graph (DAG). There are $d$ input nodes in the graph, all with an in-degree of $0$. Other nodes are classified as logic gates and are assigned one of the labels* AND, OR, *or* NOT. *We use $|C_d|$ to represent the size of $C_d$, referring to the count of nodes in the Boolean circuit.*

Therefore, we can proceed to define the languages recognizable by certain families of Boolean circuits, considering their structural constraints, gate types, and depth. These factors determine the computational power of the circuits in each family.

**Definition 3.2** (Language, Definition 6.2 in (Arora & Barak, 2009)). *Let $L \subseteq \{0, 1\}^*$ denote a language. $L$ can be recognized by a Boolean circuits family $\mathcal{C}$ if, for every string $x \in \{0, 1\}^*$, a Boolean circuit $C_{|x|} \in \mathcal{C}$ exists, which takes $x$ as input. This circuit has an input length of $|x|$, and $x \in L$ if and only if $C_{|x|}(x) = 1$ holds.*

Next, the concept of complexity classes will be given, which categorizes computational problems based on their inherent difficulty, determined by the resources—such as time or space—required to solve them. In this context, different complexity classes impose constraints on the resources of Boolean circuits, which can be further characterized by factors such as circuit size, depth, number of fan-in, and gate types. We introduce the complexity classes as the following

- A language belongs to $\mathsf{NC}^i$ class if it can be decided by a $\mathrm{poly}(d)$ size, $O(\log^i(d))$ depth boolean circuits equipped with restricted fan-in basic gates AND, OR and NOT gates.

- A language belongs to $\mathsf{AC}^i$ class if it can be decided by a $\mathrm{poly}(d)$ size, $O(\log^i(d))$ depth boolean circuits equipped with no-limit fan-in basic gates AND, OR and NOT gates.

- A language belongs to $\mathsf{TC}^i$ class if it can be decided by a $\mathrm{poly}(d)$ size, $O(\log^i(d))$ depth boolean circuits equipped with no-limit fan-in basic gates AND, OR, NOT and MAJORITY gates.

- A language belongs to $\mathsf{P}$ class if it can be decided by a deterministic Turing machine in polynomial time with respect to its input size

There is a folklore regarding the hierarchical relationships between the complexity classes mentioned above, for every $i \in \mathbb{N}$: $\mathsf{NC}^i \subseteq \mathsf{AC}^i \subseteq \mathsf{TC}^i \subseteq \mathsf{NC}^{i+1} \subseteq \mathsf{P}$. Note that the question of whether $\mathsf{TC}^0 \subsetneq \mathsf{NC}^1$ remains an open problem in circuit complexity.

In theoretical computer science, the uniformity of a complexity class refers to whether the circuit family in question can be constructed by a uniform algorithm, i.e., an algorithm that outputs a description of the circuit for any input size. Specifically, L-uniformty requires a Turing machine that uses $O(\log(d))$ space to output a circuit $C$ which can recognize a given language $L \subseteq \{0, 1\}^*$. Moreover, DLOGTIME-uniformity stipulates that a random access Turing machine must produce a circuit $C$ that recognizes a given language $L \subseteq \{0, 1\}^*$. Except in the case of small circuit complexity classes, where circuits are incapable of simulating the machines that create them, DLOGTIME-uniformity is the same as L-uniformity. For further discussion on various notions of uniformity, see (Barrington & Immerman, 1994; Hesse et al., 2002).

Throughout this work, any reference to a uniform $\mathsf{TC}^0$ should be understood as referring to a DLOGTIME-uniform $\mathsf{TC}^0$.

## 3.2 Basic Tools

In this section, we first define floating-point numbers and then illustrate a series of operations involving them. Finally, we analyze the circuit complexity associated with these operations, which is essential in the later proof.

**Definition 3.3** (Floating point number, Definition 9 in (Chiang, 2024)). *Let $p$ be an integer representing precision. Let $m \in (-2^p, -2^{p-1}] \cup \{0\} \cup [2^{p-1}, 2^p)$ denote an integer called the significance. Let $e \in [-2^p, 2^p)$ denote an integer called the exponent. A floating point number with $p$-bits is composed of the parts $m$ and $e$, and its value is given by $m \cdot 2^e$. Throughout this paper, the set of all $p$-bit floating-point numbers is denoted by $\mathsf{F}_p$.*

Then, we move forward to define the round operation of float point numbers.

**Definition 3.4** (Rounding Operation, Definition 9 in (Chiang, 2024)). *Given a floating point number $x$, we use $\mathrm{round}_p(x)$ to denote the nearest number to $x$ which is $p$-bit floating-point.*

For the definitions of addition, multiplication, division, comparison, and floor operations on floating-point numbers as outlined in Definition 3.3, refer to (Chiang, 2024). In this paper, we introduce the corresponding circuit complexity classes to which these operations belong.

**Lemma 3.5** (Operations on floating point numbers in $\mathsf{TC}^0$, Lemma 10 and Lemma 11 of (Chiang, 2024)). *Assume the precision $p \leq \mathrm{poly}(n)$. Then we have:*

- *Part 1. Given two $p$-bits float point numbers $x_1$ and $x_2$. Let the addition, division, and multiplication operations of $x_1$ and $x_2$ be outlined in (Chiang, 2024). Then, these operations can be simulated by a size bounded by $\mathrm{poly}(n)$ and constant depth bounded by $d_{\mathrm{std}}$ DLOGTIME-uniform threshold circuit.*

- *Part 2. Given $n$ $p$-bits float point number $x_1, \ldots, x_n$. The iterated multiplication of $x_1, x_2 \ldots, x_n$ can be simulated by a size bounded by $\mathrm{poly}(n)$ and constant depth bounded by $d_\otimes$ DLOGTIME-uniform threshold circuit.*

- *Part 3. Given $n$ $p$-bits float point number $x_1, \ldots, x_n$. The iterated addition of $x_1, x_2 \ldots, x_n$ can be simulated by a size bounded by $\mathrm{poly}(n)$ and constant depth bounded by $d_\oplus$ DLOGTIME-uniform threshold circuit. To be noticed, there is a rounding operation after the the summation is completed.*

Then, we show a lemma stating that we can use a $\mathsf{TC}^0$ circuit to simulate the approximated exponential function.

**Lemma 3.6** (Approximating the Exponential Operation in $\mathsf{TC}^0$, Lemma 12 of (Chiang, 2024)). *Assume the precision $p \leq \mathrm{poly}(n)$. Given any number $x$ with $p$-bit float point, the $\exp(x)$ function can be approximated by a uniform threshold circuit. This circuit has a size bounded by $\mathrm{poly}(n)$ and a constant depth $d_{\exp}$, and it guarantees a relative error of at most $2^{-p}$.*

## 4 Model Formulation

Section 4.1 presents the overall architecture of the VAR model and divides its processing workflow into three stages. In Section 4.2, we provide the mathematical formulation for the modules involved in the pyramid-shaped token map generation stage. Section 4.3 offers the mathematical formulation for the modules in the feature map reconstruction stage, while Section 4.4 presents the mathematical formulation for the modules in the VQ-VAE Decoder process stage.

### 4.1 Overall Architecture

In this section, We present the overall architecture of the VAR model and divide its processing workflow into three stages.

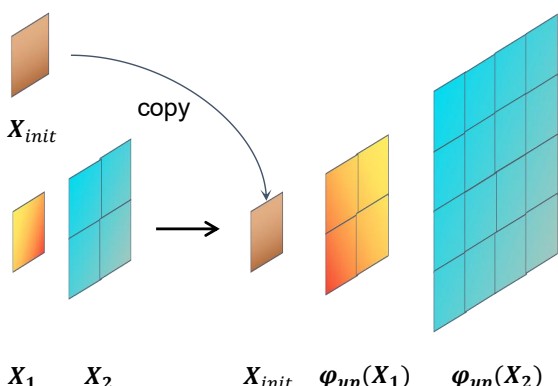

Figure 1: Example of the Pyramid Up-Interpolation Layer $\Phi_{\mathrm{up},2}$ used in the model.

**Stage 1: Pyramid-Shaped Token Maps Generation.** Firstly, the VAR model will start by quantizing an initial input token map $X_{\mathrm{init}} \in \mathbb{R}^{1 \times 1 \times d}$ into $K$ multiple scale pyramid-shaped token maps $(r_1, \ldots, r_K)$, each at an increasingly higher resolution $h_k \times w_k$. During the $k$-th autoregressive step, all the $h_k \times w_k$ will be generated in parallel, conditioned on $r_k$'s prefix $r_1, \ldots, r_{k-1}$. In Section 4.2, we provide a mathematical definition for each module in this stage.

**Stage 2: Feature Map Reconstruction.** The second stage of the VAR model is to reconstruct the generated pyramid-shaped token maps $r_1, \ldots, r_K$ into a Feature Map. Specifically, the VAR model uses an up-interpolation layer to interpolate each of the token maps $(r_1, ..., r_{K-1})$ to the size of $r_K$ and applies a convolution layer to reduce the loss introduced by the interpolation. After this process, the VAR model sums the K token maps to obtain the desired Feature Map. In Section 4.3, we provide a mathematical definition for each module in this stage.

**Stage 3: Generating Image Using VQ-VAE Decoder.** The third stage of VAR model is to use VQ-VAE Decoder to generate the final output image by taking the input of feature map. We follow the implementation of (Tian et al., 2024) and regard the VQ-VAE Decoder as a module composed of fixed-depth ResNet layers, attention layers, and up-interpolation layers. In Section 4.4, we provide a mathematical definition for each module in this stage.

### 4.2 STAGE 1: TOKEN MAPS GENERATION

The VAR model uses the VAR Transformer to convert the initialized token map $X_{\mathrm{init}}$ into a series of pyramid-shaped token maps. The VAR Transformer alternates between up sample blocks and attention layers to get the output.

**Up Sample Blocks.** The $k$-th up sample block takes as input the initial token map $X_{\mathrm{init}}$ and the previous pyramid-shaped token maps $X_1, \ldots, X_k$, sets $Y_1 = X_{\mathrm{init}}$ and up samples each $X_i$ into a new token map $Y_{i+1}$, and outputs the new pyramid-shaped token maps $Y_1, \ldots, Y_{k+1}$.

The upsampling on each token map $X_r (r \in [k])$ uses interpolation with a bicubic spline kernel.

**Definition 4.1** (Bicubic Spline Kernel). *A bicubic spline kernel is a piecewise cubic function $W : \mathbb{R} \to \mathbb{R}$ that satisfies $W(x) \in [0, 1]$ for all $x \in \mathbb{R}$.*

**Definition 4.2** (Up-interpolation Layer for One-Step Geometric Sequence). *The layer $\phi_{\mathrm{up},r}$ takes the input feature map $X_r \in \mathbb{R}^{h_r \times w_r \times d}$ and computes the output feature map $Y_{r+1} \in \mathbb{R}^{h_{r+1} \times w_{r+1} \times d}$, where $h_r < h_{r+1}$ are the heights, $w_r < w_{r+1}$ are the widths, and $d \in \mathbb{N}$ is the hidden dimension. It computes $Y_{r+1} = \phi_{\mathrm{up},r}(X_r)$ with a bicubic spline kernel $W$: for $i \in [h_{r+1}], j \in [w_{r+1}], l \in [d]$,*

$$[Y_{r+1}]_{i,j,l} := \sum_{s=-1}^{2} \sum_{t=-1}^{2} W(s) \cdot [X_r]_{\frac{i \cdot h_r}{h_{r+1}}+s, \frac{j \cdot w_r}{w_{r+1}}+t, l} \cdot W(t) \qquad (1)$$

We are now ready to present the up sample block $\Phi$.

**Definition 4.3** (Pyramid Up-Interpolation Layer $\Phi$). *The layer $\Phi_{\mathrm{up},k}$ takes the initial token map $X_{\mathrm{init}}$ and the token maps $X_r \in \mathbb{R}^{h_r \times w_r \times c}(r \in [k])$ and computes new token maps $Y_r \in \mathbb{R}^{h_r \times w_r \times c}$. It sets $Y_1 = X_{\mathrm{init}}$ and computes $Y_{r+1} = \phi_{\mathrm{up},r}(X_r)$ as in Definition 4.2. The output is the set consisting $Y_i(i \in [k+1])$.*

**Attention Layer.** After an up sample block, the token maps (after being flattened into a proper shape) will be input into an attention layer.

**Definition 4.4** (Single Attention Layer). *Let $X \in \mathbb{R}^{n \times d}$ denote the input matrix. Let $W_Q, W_K, W_V \in \mathbb{R}^{d \times d}$ denote the weight matrix for query, key, and value, respectively. First, compute the attention matrix $A \in \mathbb{R}^{n \times n}$:*

$$A_{i,j} := \exp(X_{i,*} W_Q W_K^\top X_{j,*}^\top), \ \ for \ i,j \in [n].$$

*Then, compute the output:* $\mathsf{Attn}(X) := D^{-1} A X W_V$, *where* $D := \mathrm{diag}(A \mathbf{1}_n) \in \mathbb{R}^{n \times n}$

Then, we move forward to define the multilayer perceptron layer.

**Definition 4.5** (Multilayer Perceptron layer). *Given an input matrix $X \in \mathsf{F}_p^{n \times d}$. Let $i \in [n]$. We use $g^{\mathrm{MLP}}$ to denote the MLP layer. Specifically, we have*

$$g^{\mathrm{MLP}}(X)_{i,*} := W \cdot X_{i,*} + b.$$

We then proceed to define the layerwise normalization layer.

**Definition 4.6** (Layer-wise normalization layer). *Given an input matrix $X \in \mathsf{F}_p^{n \times d}$. Let $i \in [n]$. We use $g^{\mathrm{LN}}$ to denote the LN layer. Specifically, we have*

$$g^{\mathrm{LN}}(X)_{i,*} := \frac{X_{i,*} - \mu_i}{\sqrt{\sigma_i^2}},$$

*where $\mu_i := \sum_{j=1}^d X_{i,j}/d$, and $\sigma_i^2 := \sum_{j=1}^d (X_{i,j} - \mu_i)^2/d$.*

VAR **Transformer.** A VAR Transformer with $K$ layers alternates between the attention layer and up sample blocks (where the output of each layer is reshaped to a proper shape as the input for the next layer):

**Definition 4.7** (VAR transformer). *The transformer $\mathsf{TF}$ takes an initial token map $X_{\mathrm{init}} \in \mathbb{R}^{1 \times d}$, computes $Z_0 = X_{\mathrm{init}}$, $Z_k = \Phi_{\mathrm{up},k}(X_{\mathrm{init}}, \mathsf{Attn}_k(Z_{k-1}))$, for $k \in [K-1]$ and finally outputs $\mathsf{Attn}_K(Z_{K-1})$. Here $\Phi_{\mathrm{up},k}$ is defined in Definition 4.3, $\mathsf{Attn}_i$ is defined in Definition 4.4, $Z_{k-1}$ is flatten into shape $(\sum_{r=1}^k h_r w_r) \times d$ as input for $\mathsf{Attn}_k$, and the output of $\mathsf{Attn}_k$ is reshaped into $X_r \in \mathbb{R}^{h_r \times w_r \times c}(r \in [k])$ as input for $\Phi_{\mathrm{up},k}$.*

*For convenience, we often abuse notation slightly and write:*

$$\mathsf{TF}(X_{\mathrm{init}}) := \mathsf{Attn}_K \circ \Phi_{\mathrm{up},K-1} \circ \cdots \circ \Phi_{\mathrm{up},1} \circ \mathsf{Attn}_1(X_{\mathrm{init}}),$$

*where $\circ$ denotes function composition.*

### 4.3 STAGE 2: FEATURE MAP RECONSTRUCTION

In phase 2, the VAR model will transform the generated pyramid-shaped token maps into feature maps. This phase has the following main modules:

**Up Sample Blocks.** The VAR model performs up-sampling on token maps of different sizes, scaling them to the size of the final output feature map. In this process, the VAR model will use the up-interpolation blocks defined in Definition 4.2. To mitigate information loss during token map up-scaling, the VAR model employs convolution blocks to post-process the up-scaled token maps. We define the convolution layers as the following:

**Definition 4.8** (Convolution Layer). *Let $h, w \in \mathbb{N}$ denote the height and width of the input and output feature map, respectively. Let $c_{\mathrm{in}}, c_{\mathrm{out}} \in \mathbb{N}$ denote the number of channels of the input feature map and the output feature map, respectively. Let $X \in \mathbb{R}_p^{h \times w \times c_{\mathrm{in}}}$ represent the input feature map. For $l \in [c_{\mathrm{out}}]$, we use $K^l \in \mathsf{F}_p^{3 \times 3 \times c_{\mathrm{in}}}$ to denote the $l$-th convolution kernel. Let $p = 1$ denote*

*the padding of the convolution layer. Let $s = 1$ denote the stride of the convolution kernel. Let $Y \in \mathbb{R}_p^{h \times w \times c_{\text{out}}}$ represent the output feature map. Then we use $\phi_{\text{conv}} : \mathbb{R}_p^{h \times w \times c_{\text{in}}} \to \mathbb{R}_p^{h \times w \times c_{\text{out}}}$ to represent the convolution operation then we have $Y = \phi_{\text{conv}}(X)$. Specifically, for $i \in [h], j \in [w], l \in [c_{\text{out}}]$, we have*

$$Y_{i,j,l} := \sum_{m=1}^{3} \sum_{n=1}^{3} \sum_{c=1}^{c_{\text{in}}} X_{i+m-2, j+n-2, c} \cdot K_{m,n,c}^l + b$$

**Remark 4.9.** *Assumptions of kernel size, padding of the convolution layer, and stride of the convolution kernel are based on the specific implementation of (Tian et al., 2024).*

### 4.4 STAGE 3: VQ-VAE DECODER PROCESS

VAR will use the VQ-VAE Decoder Module to reconstruct the feature map generated in Section 4.3 into a new image. The Decoder of VQ-VAE has the following main modules:

**ResNet Layers.** In the VQVAE decoder, the ResNet block, which includes two (or more) convolution blocks, plays a crucial role in improving the model's ability to reconstruct high-quality outputs. The convolution blocks help capture spatial hierarchies and patterns in the data, while the residual connections facilitate better gradient flow and allow the model to focus on learning the residuals (differences) between the input and output. The definition of convolution block is given in Definition 4.8.

**Attention Layers.** The Attention block helps the Decoder fuse information from different locations during the generation process, which can significantly improve the clarity and detail of the generated images. When applied to a feature map, the attention mechanism computes attention scores for all pairs of pixels, capturing their pairwise relationships and dependencies. The definitions of blocks in attention are given in Section 4.2.

**Up Sample Layers.** The VQ-VAE decoder uses Up-Sample Blocks to progressively increase the spatial resolution of the latent representation. The Up-Sample Blocks in VQVAE combine up-interpolation and convolution blocks to restore the spatial dimensions of the feature maps, facilitating the reconstruction of the high-resolution output image. The convolution block has already been defined in Definition 4.8, and the up-interpolation block has already been defined in Definition 4.2.

## 5 COMPLEXITY OF VAR MODELS

In this section, we present the critical findings on the circuit complexity of crucial operations in the computation of VAR models.

### 5.1 COMPUTING UP INTERPOLATION BLOCKS

In this section, we firstly show that the up-interpolation layer $\phi_{\text{up},r}$ defined in Definition 4.2 can be computed in $\mathsf{TC}^0$.

**Lemma 5.1** (Up-Interpolation Layer for One-Step Geometric Sequence belongs to $\mathsf{TC}^0$ class, informal version of Lemma B.1). *Let $m \in \mathbb{N}$ denote the number of attention layers in VAR transformer. Let $r \in [m-1]$. Let $d > 0$ denote one positive integer. Let $X_{\text{init}} \in \mathsf{F}_p^{1 \times d}$ denote the initial token map. Let $\phi_{\text{up},r} : \mathsf{F}_p^{h_r \times w_r \times d} \to \mathsf{F}_p^{h_{r+1} \times w_{r+1} \times d}$ be defined in Definition 4.2. Let $h_{r+1}$ represent the height of the token map output by $\phi_{\text{up},r}$. Let $w_{r+1}$ represent the width of the token map output by $\phi_{\text{up},r}$. Assume $h_m \leq \text{poly}(d)$ and $w_m \leq \text{poly}(d)$. Assume the precision $p \leq \text{poly}(d)$. Then we can simulate the $\phi_{\text{up},r}$ by a uniform threshold circuit with $\text{poly}(d)$ size and constant depth $O(1)$.*

**Lemma 5.2** (Pyramid Up-Interpolation Layer belongs to $\mathsf{TC}^0$ class, informal version of Lemma B.2). *Let $m \in \mathbb{N}$ denote the number of attention layers in VAR transformer. Let $r \in [m-1]$. Let $d > 0$ denote one positive integer. Let $X_{\text{init}} \in \mathsf{F}_p^{1 \times d}$ denote the initial token map. Let $\Phi_{\text{up},r} : \mathsf{F}_p^{h_{[r]} \times w_{[r]} \times d} \to \mathsf{F}_p^{h_{[r+1]} \times w_{[r+1]} \times d}$ be defined in Definition 4.3. Assume $h_m = \text{poly}(d)$ and $w_m = \text{poly}(d)$. Assume the precision $p \leq \text{poly}(d)$. Assume $m = O(1)$.*

*Then, we can simulate $\Phi_{\text{up},r}$ by a uniform threshold circuit with size bounded by $\text{poly}(d)$ and depth $O(1)$.*

## 5.2 COMPUTING ATTENTION MATRIX

Let us begin by recalling that the matrix multiplication of two matrices belongs to $\mathsf{TC}^0$.

**Lemma 5.3** (Matrix Multiplication belongs to $\mathsf{TC}^0$ class, Lemma 4.2 in (Chen et al., 2024))**.** *Assume the precision $p \leq \mathrm{poly}(d)$ and $n_1, n_2 \leq \mathrm{poly}(d)$. Let $A \in \mathbb{F}_p^{n_1 \times d}$ and $B \in \mathbb{F}_p^{d \times n_2}$. Then we can apply a* DLOGTIME-*uniform threshold circuit with constant depth $(d_{\mathrm{std}} + d_\oplus)$ and size bounded by $\mathrm{poly}(d)$ to get the matrix product $AB$.*

## 5.3 COMPUTING SINGLE ATTENTION LAYER

Subsequently, matrix operations can be applied to compute the attention matrix.

**Lemma 5.4** (Attention matrix computation belongs to $\mathsf{TC}^0$ class, informal version of Lemma B.3)**.** *Let $m \in \mathbb{N}$ denote the number of attention layers in* VAR *transformer. Let $r \in [m]$. Assume the precision $p \leq \mathrm{poly}(d)$. Let $d > 0$ denote one positive integer. Let $X_{\mathrm{init}} \in \mathsf{F}_p^{1 \times d}$ denote the initial token map. Let $\mathsf{Attn}_r$ denote the $r$-th attention layer in* VAR *transformer defined in Definition 4.4. Let $X_r \in \mathsf{F}_p^{n_r \times d}$ denote the input of $\mathsf{Attn}_r$. Let $W_Q, W_K \in \mathsf{F}_p^{d \times d}$ denote two weight matrix. Assume $h_m \leq \mathrm{poly}(d)$ and $w_m \leq \mathrm{poly}(d)$. Assume $m = O(1)$.*

*Then we can use a size bounded by $\mathrm{poly}(d)$ and constant depth $3(d_{\mathrm{std}} + d_\oplus) + d_{\exp}$ uniform threshold circuit to compute the attention matrix $A$ defined in Definition 4.4.*

Then, we analyze the complete attention layer.

**Lemma 5.5** (Single Attention Layer computation in $\mathsf{TC}^0$, informal version of Lemma B.4)**.** *Let $m \in \mathbb{N}$ denote the number of attention layers in* VAR *transformer. Let $r \in [m]$. Assume the precision $p \leq \mathrm{poly}(d)$. Let $d > 0$ denote one positive integer. Let $X_{\mathrm{init}} \in \mathsf{F}_p^{1 \times d}$ denote the initial token map. Let $\mathsf{Attn}_r$ denote the $r$-th attention layer in* VAR *transformer. Assume $h_m \leq \mathrm{poly}(d)$ and $w_m \leq \mathrm{poly}(d)$. Assume $m = O(1)$.*

*Then we can use a uniform threshold circuit with size bounded by $\mathrm{poly}(d)$ and constant depth $6(d_{\mathrm{std}} + d_\oplus) + d_{\exp}$ to simulate the attention layer $\mathsf{Attn}_r$ defined in Definition 4.4.*

## 5.4 COMPUTING COMMON COMPONENTS LAYERS

This section outlines the MLP layer circuit complexity.

**Lemma 5.6** (MLP computation falls within $\mathsf{TC}^0$ class, Lemma 4.5 of (Chen et al., 2025b))**.** *Assume the precision $p \leq \mathrm{poly}(d)$. Let $X_{\mathrm{init}} \in \mathsf{F}_p^{1 \times d}$ denote the initial token map. Then, we can use a size bounded by $\mathrm{poly}(d)$ and constant depth $2d_{\mathrm{std}} + d_\oplus$ uniform threshold circuit to simulate the MLP layer in Definition 4.5.*

Next, we examine the layer-normalization (LN) layer circuit complexity.

**Lemma 5.7** (LN computation falls within $\mathsf{TC}^0$ class, Lemma 4.6 of (Chen et al., 2025b))**.** *Assume the precision $p \leq \mathrm{poly}(d)$. Let $X_{\mathrm{init}} \in \mathsf{F}_p^{1 \times d}$ denote the initial token map. Then we can use a size bounded by $\mathrm{poly}(d)$ and constant depth $5d_{\mathrm{std}} + 2d_\oplus + d_{\mathrm{sqrt}}$ uniform threshold circuit to simulate the Layer-wise Normalization layer defined in Definition 4.6.*

## 5.5 COMPUTING CONVOLUTION BLOCKS

We prove in this section that the convolution layers can be computed within $\mathsf{TC}^0$.

**Lemma 5.8** (One Kernel Convolution Process in $\mathsf{TC}^0$, informal version of Lemma B.5)**.** *Let $d > 0$ denote one positive integer. Let $X_{\mathrm{init}} \in \mathsf{F}_p^{1 \times d}$ denote the initial token map. Let $X \in \mathsf{F}_p^{h \times w \times c_{\mathrm{in}}}$ denote the origin feature map. Let $Y \in \mathsf{F}_p^{h \times w \times c_{\mathrm{out}}}$ denote the output feature map. Assume $h, w, c_{\mathrm{in}} \leq \mathrm{poly}(d)$.*

*Then, we can apply a size bounded by $\mathrm{poly}(n)$ and $O(1)$ depth uniform threshold circuit to simulate one kernel convolution process.*

**Proposition 5.9** (Multiple Kernel Convolution Process in $\mathsf{TC}^0$)**.** *Let $d > 0$ denote one positive integer. Let $X_{\mathrm{init}} \in \mathsf{F}_p^{1 \times d}$ denote the initial token map. Assume we have $k$ convolution kernel in a*

*convolution block. Let $k \leq \text{poly}(d)$. Since the computations of different convolutional kernels can be parallelizable, then we can apply a size $\text{poly}(d)$ and $O(1)$ depth to simulate the whole process.*

*Proof.* This is can be easily derived from Lemma 5.8 and $k \leq \text{poly}(d)$. □

### 5.6 COMPUTING PHASE 1: VAR TRANSFORMER

In this part, we establish that the VAR Transformer defined in Definition 4.7 is within the computational power of $\mathsf{TC}^0$

**Lemma 5.10** (VAR Transformer computation in $\mathsf{TC}^0$, informal version of Lemma B.6). *Let $d > 0$ denote one positive integer. Let $X_{\text{init}} \in \mathsf{F}_p^{1 \times d}$ denote the initial token map. Assume the number of attention layers $m = O(1)$. Assume the precision $p \leq \text{poly}(d)$. Then, we can apply a uniform threshold circuit with size $\text{poly}(d)$ and depth $O(1)$ to simulate the* VAR *Transformer* TF *defined in Definition 4.7.*

### 5.7 COMPUTING PHASE 2: FEATURE MAP RECONSTRUCTION

In this section, we show that the feature map reconstruction is within the computational power of $\mathsf{TC}^0$.

**Lemma 5.11** (Feature Map Reconstruction computation in $\mathsf{TC}^0$.). *Let $d > 0$ denote one positive integer. Let $X_{\text{init}} \in \mathsf{F}_p^{1 \times d}$ denote the initial token map. Assume the number of the up-interpolation layers and convolutional layers in the Feature Map Reconstruction phase is constant $O(1)$. Assuming the precision $p \leq \text{poly}(d)$, then we can apply a uniform threshold circuit with size $\text{poly}(d)$ and $O(1)$ depth to simulate the feature map reconstruction operations.*

*Proof.* This can be easily derived from Lemma 5.1 and Proposition 5.9. □

### 5.8 COMPUTING PHASE 3: VQ-VAE DECODER PROCESS

In this section, we show that the VQ-VAE Decoder is within the computational power of $\mathsf{TC}^0$

**Lemma 5.12** (VQ-VAE Decoder process in $\mathsf{TC}^0$, informal version of Lemma B.7). *Let $d > 0$ denote one positive integer. Let $X_{\text{init}} \in \mathsf{F}_p^{1 \times d}$ denote the initial token map. Assume the precision $p \leq \text{poly}(d)$. Assume the number of the ResNet layers, attention layers, and Up-Sample layers in VQ-VAE Decoder is constant $O(1)$. Then, we can apply a uniform threshold circuit with size $\text{poly}(d)$ and $O(1)$ depth to simulate the VQ-VAE decoder process.*

### 5.9 MAIN RESULT

We present our main result, which derives the circuit complexity limits for the VAR model.

**Theorem 5.13** (Circuit complexity of the VAR model.). *Let $d > 0$ denote one positive integer. Let $X_{\text{init}} \in \mathsf{F}_p^{1 \times d}$ denote the initial token map. Assuming precision $p \leq \text{poly}(d)$, then we can apply a uniform threshold circuit to simulate the* VAR *model, where the circuit has size $\text{poly}(d)$ and $O(1)$ depth.*

*Proof.* This result directly comes from Lemma 5.10, Lemma 5.11 and Lemma 5.12. □

## 6 CONCLUSION

This study provides a comprehensive theoretical analysis of VAR models, deriving key limits on their computational abilities. Our approach centers on examining the circuit complexity of various components of VAR models, from the up-interpolation layers and the convolution layers to the attention mechanism. Furthermore, we show that VAR can be expressed as uniform $\mathsf{TC}^0$ circuits. This finding is important because it exposes inherent constraints in the expressiveness of VAR models, despite their empirical effectiveness.

## ETHIC STATEMENT

This paper does not involve human subjects, personally identifiable data, or sensitive applications. We do not foresee direct ethical risks. We follow the ICLR Code of Ethics and affirm that all aspects of this research comply with the principles of fairness, transparency, and integrity.

## REPRODUCIBILITY STATEMENT

We ensure reproducibility of our theoretical results by including all formal assumptions, definitions, and complete proofs in the appendix. The main text states each theorem clearly and refers to the detailed proofs. No external data or software is required.

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

# Appendix

**Roadmap.** In Section A, we introduce the notations we used in appendix. In Section B, we introduce the missing proofs in Section B.

## A    NOTATIONS

We apply $[n]$ to represent the set $\{1, 2, \cdots, n\}$ for any positive integer $n$. The set of natural numbers is denoted by $\mathbb{N} := \{0, 1, 2, \ldots\}$. Let $X \in \mathbb{R}^{m \times n}$ be a matrix, where $X_{i,j}$ refers to the element at the $i$-th row and $j$-th column. When $x_i$ belongs to $\{0, 1\}^*$, it signifies a binary number with arbitrary length. In a general setting, $x_i$ represents a length $p$ binary string, with each bit taking a value of either 0 or 1.

## B    MISSING PROOFS IN SECTION 5

In this section, we present some missing proofs in Section 5.

**Lemma B.1** (Up-Interpolation Layer for One-Step Geometric Sequence belongs to $\mathsf{TC}^0$ class, formal version of Lemma 5.1). *If the following conditions hold:*

- *Let $m \in \mathbb{N}$ denote the number of attention layers in* VAR *transformer.*

- *Let $r \in [m-1]$.*

- *Let $d > 0$ denote one positive integer.*

- *Let $X_{\mathrm{init}} \in \mathsf{F}_p^{1 \times d}$ denote the initial token map.*

- *Let $\phi_{\mathrm{up},r} : \mathsf{F}_p^{h_r \times w_r \times d} \to \mathsf{F}_p^{h_{r+1} \times w_{r+1} \times d}$ be defined in Definition 4.2.*

- *Let $h_{r+1}$ represent the height of the token map output by $\phi_{\mathrm{up},r}$.*

- *Let $w_{r+1}$ represent the width of the token map output by $\phi_{\mathrm{up},r}$.*

- *Assume $h_m \leq \mathrm{poly}(d)$ and $w_m \leq \mathrm{poly}(d)$.*

- *Assume the precision $p \leq \mathrm{poly}(d)$.*

*Then we can simulate the $\phi_{\mathrm{up},r}$ by a uniform threshold circuit with $\mathrm{poly}(d)$ size and constant depth $O(1)$.*

*Proof.* Firstly, we consider the computation of each entry in the output token map. For $i \in [h_{r+1}], j \in [w_{r+1}], l \in [d]$, we have

$$Y_{i,j,l} = \sum_{s=-1}^{2} \sum_{t=-1}^{2} W(s) \cdot X_{\frac{ih}{h'}+s, \frac{jw}{w'}+t, q} \cdot W(t)$$

By using the result of Part 1 of Lemma 3.5, we can apply a uniform threshold circuit with constant depth $2d_{\mathrm{std}}$ and size bounded by $\mathrm{poly}(d)$ to compute each product $W(s) \cdot X_{\frac{ih}{h'}+s, \frac{jw}{w'}+t, q} \cdot W(t)$. Since the products for different $s$ and $t$ can be parallel computed, the uniform threshold circuit's depth for all products $W(u) \cdot X_{\frac{ih}{h'}+s, \frac{jw}{w'}+t, q}$ stays $2d_{\mathrm{std}}$.

Then, by using the result of Part 3 in Lemma 3.5, we can use a uniform threshold circuit with depth $d_{\oplus}$ and size bounded by $\mathrm{poly}(d)$ to model the sum operation:

$$\sum_{s=-1}^{2} \sum_{t=-1}^{2} W(s) \cdot X_{\frac{ih}{h'}+s, \frac{jw}{w'}+t, q} \cdot W(t)$$

Hence, we already know that the computation of one entry in the ourput token map can be simulated by a uniform threshold circuit with depth $2d_{\mathrm{std}} + d_{\oplus}$ and size bounded by $\mathrm{poly}(d)$. Since we can parallel compute $Y_{i,j,l}$ for all $i \in [h_{r+1}], j \in [w_{r+1}], l \in [d]$. So the total depth of the uniform threshold circuit still remains $2d_{\mathrm{std}} + d_{\oplus}$, and the total size of the uniform threshold circuit still remains $\mathrm{poly}(d)$ which is due to the condition that $h_{r+1} \leq h_m = \mathrm{poly}(d), w_{r+1} \leq w_m = \mathrm{poly}(d)$.

Thus we complete the proof.

$\square$

**Lemma B.2** (Pyramid Up-Interpolation Layer belongs to $\mathsf{TC}^0$ class, formal version of Lemma 5.2)**.** *If the following conditions hold:*

- *Let $m \in \mathbb{N}$ denote the number of attention layers in* VAR *transformer.*

- *Let $r \in [m-1]$.*

- *Let $d > 0$ denote one positive integer.*

- *Let $X_{\mathrm{init}} \in \mathsf{F}_p^{1 \times d}$ denote the initial token map.*

- *Let $\Phi_{\mathrm{up},r} : \mathsf{F}_p^{h_{[r]} \times w_{[r]} \times d} \to \mathsf{F}_p^{h_{[r+1]} \times w_{[r+1]} \times d}$ be defined in Definition 4.3.*

- *Assume $h_m = \mathrm{poly}(d)$ and $w_m = \mathrm{poly}(d)$.*

- *Assume the precision $p \leq \mathrm{poly}(d)$.*

- *Assume $m = O(1)$.*

*Then we can simulate $\Phi_{\mathrm{up},r}$ by a uniform threshold circuit with size bounded by $\mathrm{poly}(d)$ and depth $O(1)$.*

*Proof.* By Definition 4.3, we know that $\Phi_{\mathrm{up},r}$ is composed of $r$ layers $\phi_{\mathrm{up},i}$ where $i \in [r]$. Since for every $i \in [r]$, we can use a uniform threshold circuit with size bounded by $\mathrm{poly}(d)$ and depth $O(1)$ to simulate $\phi_{\mathrm{up},i}$ which is due to Lemma 5.1. Then we can derive that we can use a uniform threshold circuit with size bounded by $\mathrm{poly}(d)$ and depth $O(1)$ to simulate $\Phi_{\mathrm{up},r}$ which is due to chain these $r$ uniform threshold circuits together, and $r \leq m = O(1)$. $\square$

Then we move forward to present the proof of Lemma 5.4.

**Lemma B.3** (Attention matrix computation belongs to $\mathsf{TC}^0$ class, formal version of Lemma 5.4)**.** *If the following conditions hold:*

- *Let $m \in \mathbb{N}$ denote the number of attention layers in* VAR *transformer.*

- *Let $r \in [m]$*

- *Assume the precision $p \leq \mathrm{poly}(d)$.*

- *Let $d > 0$ denote one positive integer.*

- *Let $X_{\mathrm{init}} \in \mathsf{F}_p^{1 \times d}$ denote the initial token map.*

- *Let $\mathsf{Attn}_r$ denote the $r$-th attention layer in* VAR *transformer defined in Definition 4.4.*

- *Let $X_r \in \mathsf{F}_p^{n_r \times d}$ denote the input of $\mathsf{Attn}_r$.*

- *Let $W_Q, W_K \in \mathsf{F}_p^{d \times d}$ denote two weight matrix.*

- *Assume $h_m \leq \mathrm{poly}(d)$ and $w_m \leq \mathrm{poly}(d)$.*

- *Assume $m = O(1)$.*

*Then we can use a size bounded by $\mathrm{poly}(d)$ and constant depth $3(d_{\mathrm{std}} + d_{\oplus}) + d_{\exp}$ uniform threshold circuit to compute the attention matrix $A$ defined in Definition 4.4.*

*Proof.* By Definition 4.7 and Definition 4.3, we can derive that $n_i = \sum_{i=1}^{r} h_i w_i$. Since for every $i \in [r]$, we have $h_i \leq h_m \leq \mathrm{poly}(d)$ and $w_i \leq w_m \leq \mathrm{poly}(d)$, we can derive that $n_r \leq \mathrm{poly}(d)$.

Based on Lemma 5.3, we can compute the matrix product $W_Q W_K^\top$ by using a size bounded by $\mathrm{poly}(d)$ and constant depth $d_{\mathrm{std}} + d_\oplus$ uniform threshold circuit.

Then, we move forward to compute the scalar product, which is

$$t_{i,j} = X_{i,*} W_Q W_K^\top X_{j,*}^\top$$

And by using the result of Lemma 5.3, we can compute $t_{i,j}$ by applying a uniform threshold circuit, where the circuit has a polynomial-size bounded by $\mathrm{poly}(d)$ and constant depth $2(d_{\mathrm{std}} + d_\oplus)$.

In the next step, from Lemma 3.6, we can compute the exponential function $A_{i,j} = \exp(t_{i,j})$ by applying a size bounded by $\mathrm{poly}(d)$ and constant depth $d_{\exp}$ uniform threshold circuit.

After combining depths from all steps, the total depth of the circuit for computing $A_{i,j}$ is

$$d_{\mathrm{total}} = 3(d_{\mathrm{std}} + d_\oplus) + d_{\exp}.$$

Since we can parallel compute all entries in $A_{i,j}$ for $i, j \in [n_r]$, the circuit depth remains $3(d_{\mathrm{std}} + d_\oplus) + d_{\exp}$ and size bounded by $\mathrm{poly}(d)$.

Thus, we have proven the result. $\square$

Here we state the proof of Lemma 5.5.

**Lemma B.4** (Single Attention Layer computation in $\mathsf{TC}^0$, formal version of Lemma 5.5)**.** *If the following conditions hold:*

- *Let $m \in \mathbb{N}$ denote the number of attention layers in* VAR *transformer.*

- *Let $r \in [m]$*

- *Assume the precision $p \leq \mathrm{poly}(d)$.*

- *Let $d > 0$ denote one positive integer.*

- *Let $X_{\mathrm{init}} \in \mathsf{F}_p^{1 \times d}$ denote the initial token map.*

- *Let* $\mathsf{Attn}_r$ *denote the $r$-th attention layer in* VAR *transformer.*

- *Let $X_r \in \mathsf{F}_p^{n_r \times d}$ denote the input of* $\mathsf{Attn}_r$*.*

- *Let $W_V \in \mathsf{F}_p^{d \times d}$ denote a weight matrix.*

- *Assume $h_m \leq \mathrm{poly}(d)$ and $w_m \leq \mathrm{poly}(d)$.*

- *Assume $m = O(1)$.*

*Then we can use a uniform threshold circuit with size bounded by $\mathrm{poly}(d)$ and constant depth $6(d_{\mathrm{std}} + d_\oplus) + d_{\exp}$ to simulate the attention layer* $\mathsf{Attn}_r$ *defined in Definition 4.4.*

*Proof.* In Definition 4.4, we know that we need to multiply 4 matrix, namely $D^{-1}, A, X_r, W_V$. Firstly, we consider the computation of $D := \mathrm{diag}(A \mathbf{1}_{n_r})$. $D$ can be computed using a uniform threshold circuit of depth $d_\oplus$, size $\mathrm{poly}(d)$ following Part 3 of Lemma 3.5. By Lemma 5.4, computing $A$ needs a circuit of depth $3(d_{\mathrm{std}} + d_\oplus) + d_{\exp}$ and size $\mathrm{poly}(d)$. Then, we can multiply $AXW_V$, which can be computed by a depth $2(d_{\mathrm{std}} + d_\oplus)$, size $\mathrm{poly}(d)$ uniform threshold circuit following from Lemma 5.3. Finally, we can compute $D^{-1} \cdot AXW_V$ by apply division in parallel, which can be computed by a depth $d_{\mathrm{std}}$, size $\mathrm{poly}(d)$ uniform threshold circuit following from Part 1 of Lemma 3.5. Chaining above circuit, we have

$$d_{\mathrm{total}} = 6(d_{\mathrm{std}} + d_\oplus) + d_{\exp}.$$

And the size of the circuit is still $\mathrm{poly}(d)$. thus we have shown that $\mathsf{Attn}$ can be computed by a depth $6(d_{\mathrm{std}} + d_\oplus) + d_{\exp}$, size $\mathrm{poly}(d)$ uniform threshold circuit. $\square$

Then we move forward to present the proof of Lemma 5.8.

**Lemma B.5** (One Kernel Convolution Process in $\mathsf{TC}^0$, formal version of Lemma 5.8). *Under the premise that the following conditions apply:*

- *Let $d > 0$ denote one positive integer.*

- *Let $X_{\mathrm{init}} \in \mathsf{F}_p^{1 \times d}$ denote the initial token map.*

- *Let $h \in \mathbb{N}$ denote the height of the input and output feature map.*

- *Let $w \in \mathbb{N}$ denote the width of the input and output feature map.*

- *Let $c_{\mathrm{in}} \in \mathbb{N}$ denote the number of channels of the input feature map.*

- *Let $X \in \mathsf{F}_p^{h \times w \times c_{\mathrm{in}}}$ denote the origin feature map.*

- *Let $Y \in \mathsf{F}_p^{h \times w \times c_{\mathrm{out}}}$ denote the output feature map .*

- *Let $l \in [c_{\mathrm{out}}]$.*

- *Let $K^l \in \mathsf{F}_p^{3 \times 3 \times c_{\mathrm{in}}}$ denote the l-th convolution kernel.*

- *For $i \in [h]$ and $j \in [w]$.*

- *Let $h, w, c_{\mathrm{in}} \le \mathrm{poly}(d)$.*

*Then, we can apply a size bounded by $\mathrm{poly}(n)$ and $O(1)$ depth uniform threshold circuit to simulate one kernel convolution process.*

*Proof.* For each $i \in [h]$ and $j \in [w]$, we know

$$Y_{i,j,l} := \sum_{m=1}^{3} \sum_{n=1}^{3} \sum_{q=1}^{c_{\mathrm{in}}} X_{i+m-2, j+n-2, q} \cdot K_{m,n,q}^l + b$$

By using the result of Part 1 in Lemma 3.5, we can use a size bounded by $\mathrm{poly}(d)$ and $O(1)$ depth uniform threshold circuit to compute each product $X_{i+m-1, j+n-1, q} \cdot K_{m,n,q}$. Furthermore, the computation of $X_{i+m-1, j+n-1, q} \cdot K_{m,n,q}$ can be performed in parallel for all $m \in [3]$, $n \in [3]$ and $q \in [c_{\mathrm{in}}]$. Therefore, the total depth of the circuit remains $O(1)$, and its size stays $\mathrm{poly}(d)$, since $h_k \times w_k \times c \le \mathrm{poly}(d)$.

Then, we proceed to compute the sum $\sum_{m=1}^{3} \sum_{n=1}^{3} \sum_{q=1}^{c_{\mathrm{in}}} X_{i+m-2, j+n-2, q} \cdot K_{m,n,q}^l + b$. Using the result from Lemma 3.5, we can use a size bounded by $\mathrm{poly}(d)$ and $O(1)$ depth uniform threshold circuit to compute the sum. By computing $Y_{i,j}$ for all $i \in [h], j \in [w]$ in parallel, we maintain the uniform threshold circuit with $O(1)$ depth and size bounded by $\mathrm{poly}(d)$ which is due to $h, w \le \mathrm{poly}(d)$.

Thus, we can apply a size bounded by $\mathrm{poly}(d)$ and $O(1)$ depth uniform threshold circuit to simulate the one kernel convolution process. $\square$

Then we show the proof of Lemma 5.10.

**Lemma B.6** (VAR Transformer computation in $\mathsf{TC}^0$, formal version of Lemma 5.10). *Let $d > 0$ denote one positive integer. Let $X_{\mathrm{init}} \in \mathsf{F}_p^{1 \times d}$ denote the initial token map. Assume the number of attention layers $m = O(1)$. Assume the precision $p \le \mathrm{poly}(d)$. Then, we can apply a uniform threshold circuit with size $\mathrm{poly}(d)$ and depth $O(1)$ to simulate the VAR Transformer TF defined in Definition 4.7.*

*Proof.* By using the result of Lemma 5.2, we can apply a uniform threshold circuit of size bounded by $\mathrm{poly}(d)$ and $O(1)$ depth to simulate Pyramid Up-Interpolation layer $\Phi_{\mathrm{up},i}$ defined in Definition 4.3 for every $i \in [m-1]$.

By using the result of Lemma 5.5, we can apply a size bounded by $\mathrm{poly}(d)$ and $O(1)$ depth uniform threshold circuit to simulate $\mathsf{Attn}_i$ defined in Definition 4.4 for every $i \in [m]$.

By using the result of Lemma 5.6 and Lemma 5.7, we can apply a size bounded by $\mathrm{poly}(d)$ and $O(1)$ depth uniform threshold circuit to simulate $g_i$, for each $i \in [m]$.

To compute $\mathsf{TF}(X)$, we must compute $g_1, \ldots, g_m$ ,$\mathsf{Attn}_1, \ldots, \mathsf{Attn}_m$ and $\Phi_{\mathrm{up},1}, \ldots, \Phi_{\mathrm{up},m-1}$. Then, we can have that the size of the uniform threshold circuit is bounded by $\mathrm{poly}(d)$, and the total depth of the circuit is $O(1)$, which is due to $m = O(1)$.

Thus, we complete the proof. $\qquad\square$

Finally we show the proof of Lemma 5.12.

**Lemma B.7** (VQ-VAE Decoder process in $\mathsf{TC}^0$, formal version of Lemma 5.12). *Let $d > 0$ denote one positive integer. Let $X_{\mathrm{init}} \in \mathsf{F}_p^{1 \times d}$ denote the initial token map. Assume the precision $p \leq \mathrm{poly}(d)$. Assume the number of the ResNet layers, attention layers and Up-Sample layers in VQ-VAE Decoder is constant $O(1)$. Then, we can apply a uniform threshold circuit with size $\mathrm{poly}(d)$ and $O(1)$ depth to simulate the VQ-VAE decoder process.*

*Proof.* Firstly, by using the result of Proposition 5.9, we can simulate the ResNet layers by using a size $\mathrm{poly}(d)$ and $O(1)$ depth uniform threshold circuit.

Then, by using the result of Lemma 5.5, we can simulate the attention layers by using a size $\mathrm{poly}(d)$ and $O(1)$ depth uniform threshold circuit.

And, by using the result of Lemma 5.1, we can simulate the Up Sample Blocks by using a size $\mathrm{poly}(d)$ and depth $O(1)$ uniform threshold circuit.

By combing the result above, we have that a size $\mathrm{poly}(d)$ and $O(1)$ depth uniform threshold circuit can be applied to simulate the VQ-VAE decoder process. $\qquad\square$

## LLM Usage Disclosure

LLMs were used only to polish language, such as grammar and wording. These models did not contribute to idea creation or writing, and the authors take full responsibility for this paper's content.

