# OpenReview forum: "Circuit Complexity Bounds for Visual Autoregressive Model"
_ICLR.cc/2026/Conference — Submitted to ICLR 2026_

### Official Review · Reviewer_R5cA · 2025-10-22

**Soundness:** 3
**Presentation:** 2
**Contribution:** 3
**Rating:** 6
**Confidence:** 2

**Summary:**

This paper provides the first circuit-complexity analysis of Visual Autoregressive models, formally showing that a fixed-depth (O(1)), poly(d)-precision VAR—comprising bicubic up-interpolation, attention, MLP/LN, convolutions, and a constant-depth VQ-VAE decoder—can be simulated by a DLOGTIME-uniform TC0 threshold-circuit family of polynomial size and constant depth. By mathematically formulating VAR’s coarse-to-fine “next-scale” architecture and proving TC0 realizations for each component (including exp approximation and matrix ops), the work establishes a strong uniform TC0 upper bound on VAR’s expressive power, thereby clarifying inherent computational limitations and aligning VAR with recent TC0 bounds for Transformers and SSMs.

**Strengths:**

* First circuit-complexity treatment of VAR models; clean formalization of the “next-scale” autoregressive architecture by applying the known TC0 techniques (floating-point ops, exp approximation, matrix/convolution ops, attention) to a new domain.
* I'm not an expert in this domain, but the stage-wise proof structure makes the argument easy to follow; explicit statements of depth/size bounds and uniformity aid reproducibility.

**Weaknesses:**

* The main TC0 upper bound critically relies on O(1) depth and poly(d) precision/width; the paper does not analyze how bounds change when depth scales with resolution or when precision changes, limiting relevance to large practical VARs. Provide depth-parameterized bounds or discuss thresholds where the simulation may leave TC0.

**Questions:**

* Many bounds assume hm, wm ≤ poly(d). In practical VARs, d may be modest while image size grows large—how would your analysis scale if hm, wm are the primary growth parameters?

---

### Official Review · Reviewer_raET · 2025-10-28

**Soundness:** 2
**Presentation:** 1
**Contribution:** 2
**Rating:** 2
**Confidence:** 4

**Summary:**

The paper analyzes the expressiveness of visual autoregressive models using circuit complexity, formalizing VAR modules and proving they can be simulated by shallow threshold circuits under certain assumptions. The main result shows that the entire VAR pipeline can be simulated in O(1) depth and polynomial size, connecting VAR architectures to established circuit complexity classes.

**Strengths:**

- The paper attempts to formalize the components of VAR models and provides some logical structure.

- The authors reference existing circuit complexity results and applies them to the VAR setting.

**Weaknesses:**

- There is substantial overlap (e.g., Figure 1) with another ICLR submission (submission 2833), both focusing on theoretical complexity of VARs and even using exactly the same figures. This strong similarity raise concerns about originality and possible being written by LLMs.

- The contribution is mainly upper bounds. no lower bounds or separation results are provided, limiting theoretical novelty.

- Some key assumptions (e.g., constant depth/layers) in this paper do not match real-world VAR configurations, reducing practical relevance.

- Figure 1 contains unreadable symbols and overlaps. Moreover,

**Questions:**

How would the reported results change if the number of layers grows with input size, rather than being constant?

---

### Official Review · Reviewer_stB8 · 2025-10-29

**Soundness:** 2
**Presentation:** 2
**Contribution:** 2
**Rating:** 2
**Confidence:** 3

**Summary:**

This paper establishes the first circuit complexity bound for Visual AutoRegressive (VAR) models, demonstrating that they can be simulated by a DLOGTIME-uniform TC0 threshold circuit with constant depth, poly(n) size, and poly(n) precision, despite their strong empirical performance in image generation. The authors systematically analyze each component including up-interpolation blocks, attention layers, convolution blocks, and the VQ-VAE decoder to prove that all can be computed within TC0 under realistic precision assumptions.

**Strengths:**

The paper provides a technical result that extends known TC0 bounds to VAR models.

**Weaknesses:**

1. The paper largely follows existing circuit complexity analysis techniques developed for Transformers and Mamba, offering limited novel methodological or theoretical advancements.
2. While the paper claims theoretical limitations for VAR, it fails to reconcile this with its strong empirical performance, leaving the tension between theory and practice unaddressed.
3. What practical implications does the paper’s conclusion that VAR models lie within TC0 have for real-world modeling or algorithm design?
4. Figure 1 is identical to that in https://openreview.net/forum?id=S3Fq8E9jb7, raising serious concerns about originality and proper attribution.

**Questions:**

See weakness.

---

### Official Review · Reviewer_Dzt4 · 2025-10-31

**Soundness:** 3
**Presentation:** 3
**Contribution:** 3
**Rating:** 4
**Confidence:** 1

**Summary:**

This paper conducts a comprehensive theoretical investigation into the circuit complexity bounds of Visual Autoregressive (VAR) models, which adopt a coarse-to-fine "next-scale prediction" framework for image generation. The core result demonstrates that VAR models (with hidden dimension d and poly(d) precision) can be simulated by DLOGTIME-uniform $TC^0$ threshold circuits with polynomial size and constant depth. These findings reveal inherent expressive limitations of VAR models despite their empirical performance advantages.

**Strengths:**

Due to significant differences between my research domain and the circuit complexity/visual autoregressive modeling field of this paper, I am unable to provide a substantive assessment of its originality, quality, clarity, and significance from a professional perspective. The paper appears to address an underexplored gap (circuit complexity analysis of VAR models) and presents a structured theoretical framework with detailed definitions and proofs, which suggests careful academic rigor. However, a precise evaluation of whether its contributions (e.g., novel formulations, complexity bounds) are impactful or original within the field requires expertise in computational complexity that I do not possess.

**Weaknesses:**

See Strengths.

**Questions:**

No question

---

### Meta-Review · Area_Chair_anTN · 2026-01-07

**Summary:**

This paper studies the circuit complexity of Visual Autoregressive (VAR) models and proves that, under fixed depth and polynomial precision/width assumptions, VAR architectures can be simulated by DLOGTIME-uniform TC⁰ threshold circuits. Most reviewers raised significant concerns about (i) limited novelty relative to prior TC⁰ analyses of Transformers/SSMs, (ii) heavy reliance on unrealistic constant-depth and size assumptions, (iii) weak discussion of practical implications, and (iv) serious originality and presentation issues, including an allegedly duplicated figure from another submission. These concerns dominate the discussion and drive the recommendation.

**Reviewer Concerns:**

There is no rebuttal from the authors.

**Reviewer Scores:**

None will change as there is no rebuttal from the authors.

---

### Decision · Program_Chairs · 2026-01-26

Reject